# A Prediction Model for Severe Complications after Elective Colorectal Cancer Surgery in Patients of 70 Years and Older

**DOI:** 10.3390/cancers13133110

**Published:** 2021-06-22

**Authors:** Esteban T. D. Souwer, Esther Bastiaannet, Ewout W. Steyerberg, Jan Willem T. Dekker, Willem H. Steup, Marije M. Hamaker, Dirk J. A. Sonneveld, Thijs A. Burghgraef, Frederiek van den Bos, Johanna E. A. Portielje

**Affiliations:** 1Department of Internal Medicine, Haga Hospital, 2545 AA Den Haag, The Netherlands; 2Department of Medical Oncology, Leiden University Medical Center, 2333 ZA Leiden, The Netherlands; e.bastiaannet@lumc.nl (E.B.); j.e.a.portielje@lumc.nl (J.E.A.P.); 3Department of Medical Statistics, Leiden University Medical Center, 2333 ZA Leiden, The Netherlands; e.w.steyerberg@lumc.nl; 4Department of Surgery, Reinier De Graaf Gasthuis, 2625 AD Delft, The Netherlands; j.w.t.dekker@rdgg.nl; 5Department of Surgery, Haga Hospital, 2545 AA Den Haag, The Netherlands; w.steup@hagaziekenhuis.nl; 6Department of Geriatric Medicine, Diakonessenhuis, 3582 KE Utrecht, The Netherlands; mhamaker@diakonessenhuis.nl; 7Department of Surgery, Dijklander Ziekenhuis, 1624 NP Hoorn, The Netherlands; D.J.A.Sonneveld@westfriesgasthuis.nl; 8Department of Surgery, Meander Medisch Centrum, 3813 TZ Amersfoort, The Netherlands; TA.Burghgraef@meandermc.nl; 9Department of Geriatric Medicine, Leiden University Medical Center, 2333 ZA Leiden, The Netherlands; f.van_den_bos@lumc.nl

**Keywords:** colorectal cancer, surgery, frailty, prediction, postoperative complications

## Abstract

**Simple Summary:**

The objective was to develop and internally validate a predictive model based on preoperative predictors, including geriatric characteristics, for severe postoperative complications after elective surgery for stage I–III CRC in patients ≥70 years. Potential predictors included demographics, comorbidity, tumour location, activities of daily living (ADL), history of falls, malnutrition, risk factors for delirium, use of mobility aid and polypharmacy. The least absolute shrinkage and selection operator (LASSO) method was used for predictor selection and prediction model building. A geriatric model that included gender, previous DVT or pulmonary embolism, COPD/asthma/emphysema, rectal cancer, the use of a mobility aid, ADL assistance, previous delirium and polypharmacy showed satisfactory discrimination with an AUC of 0.69 (95% CI 0.73–0.64); the AUC for the optimism corrected model was 0.65. An eight-item colorectal geriatric model (GerCRC) was developed. After external validation, this risk model has the potential to be used for preoperative (shared) decision-making.

**Abstract:**

Introduction Older patients have an increased risk of morbidity and mortality after colorectal cancer (CRC) surgery. Existing CRC surgical prediction models have not incorporated geriatric predictors, limiting applicability for preoperative decision-making. The objective was to develop and internally validate a predictive model based on preoperative predictors, including geriatric characteristics, for severe postoperative complications after elective surgery for stage I–III CRC in patients ≥70 years. Patients and Methods: A prospectively collected database contained 1088 consecutive patients from five Dutch hospitals (2014–2017) with 171 severe complications (16%). The least absolute shrinkage and selection operator (LASSO) method was used for predictor selection and prediction model building. Internal validation was done using bootstrapping. Results: A geriatric model that included gender, previous DVT or pulmonary embolism, COPD/asthma/emphysema, rectal cancer, the use of a mobility aid, ADL assistance, previous delirium and polypharmacy showed satisfactory discrimination with an AUC of 0.69 (95% CI 0.73–0.64); the AUC for the optimism corrected model was 0.65. Based on these predictors, the eight-item colorectal geriatric model (GerCRC) was developed. Conclusion: The GerCRC is the first prediction model specifically developed for older patients expected to undergo CRC surgery. Combining tumour- and patient-specific predictors, including geriatric predictors, improves outcome prediction in the heterogeneous older population.

## 1. Introduction

Older patients make up the majority of newly diagnosed patients with colorectal cancer (CRC) [1], and for this heterogeneous population, risks and benefits of treatment must be weighed at an individual level [2,3,4,5]. Prediction models can be used to facilitate decision-making and estimate outcomes of treatment such as surgery-related morbidity and mortality. Especially severe complications are of interest because they hinder the postoperative course and impact postoperative functioning and quality of life of older patients [6,7,8].

For older patients with CRC, potential predictors for these outcomes include physical performance measures [9,10,11], falls and cognitive impairments [12,13]. However, in currently available prediction models, there is a focus on cancer- and surgery-related predictors. At the same time, the inclusion of perioperative predictors in many models limits their use for preoperative decision-making [14,15,16].

We have previously shown that most available CRC prediction models have a moderate to high risk of bias, especially in older adults [17]. That also applies to the three surgical risk prediction models for prediction of severe complications: the Physiology and Operative Severity Score for the Enumeration of Mortality and Morbidity (POSSUM) [14], Colorectal Biochemical and Hematological Outcome Model (CR-BHOM) [18] and the American College of Surgeons National Surgical Quality Improvement Program (ACS-NSQIP) [10,19]. Predictors related to geriatric characteristics might improve a prediction model’s performance for older CRC patients [13,20]. 

This study aimed to develop and internally validate a prognostic preoperative clinical model for severe postoperative complications after elective surgery for stage I–III CRC, intended to support shared decision-making with older patients. We analysed data from a large population-based cohort of patients ≥70 years.

## 2. Materials and Methods

### 2.1. Data and Participants

This study is reported in accordance with the recommendations set forth by The Transparent Reporting of a Multivariable Prediction Model for Individual Prognosis or Diagnosis (TRIPOD) Initiative [21]. Ethical approval was obtained the Medical Ethics Committee Southwest Holland (September 2016) and informed consent were waived due to the retrospective nature of this study. 

Five Dutch hospitals provided data for this study. Patient demographic data, as well as outcome data, were retrieved from the Dutch Colorectal Audit (DCRA) between January 2014 and December 2017. The DCRA is a national mandatory surgical database that contains pre-, peri- and postoperative surgical and outcome data on all operated CRC patients in the Netherlands as part of a national quality improvement project. From the electronic medical records (EMRs), geriatric data were retrieved that were registered as part of standard preoperative care. 

Patients who were 70 years or older on the day of surgery were identified from the DCRA. All consecutive patients are prospectively enrolled in this database by qualified staff [22]. Eligible for inclusion were patients with elective surgery for stage I–III CRC. Exclusion criteria were synchronous cancer at diagnosis and nonelective or transanal endoscopic microsurgery (TEMS). 

In all participating hospitals, both laparoscopy and Enhanced Recovery After Surgery (ERAS, anaesthesiology guidelines and postoperative care) [23] were considered the standard of care and therefore most likely used in the majority of patients. Other standard care measures and interventions during the study period were the detection of undernutrition and dietary support when needed, postoperative physiotherapy in case of functional dependency (activities of daily living (ADL)) and early detection of delirium in high-risk patients.

The risk of malnutrition is assessed preoperatively with the screening tools Short Nutritional Assessment Questionnaire (SNAQ) [24] or Malnutrition Universal Screening Tool (MUST) [25]. Additional information is collected on high-risk patients using a more comprehensive intake with additional nutritional interventions by a dietician when needed.

### 2.2. Outcome

A complication was defined as in-and-out of hospital morbidity (of any kind) within 30 days of surgery. A severe complication was defined as a complication leading to ICU admission (more than 2 days), a reintervention (surgical or radiological), prolonged hospital stay (more than 14 days) or postoperative mortality. This is consistent with previous publications in which outcome data from the DCRA were analysed [26].

### 2.3. Predictors

A systematic review of prediction models for adverse outcomes of CRC was used to identify commonly used predictors in younger and older patients [17]. Candidate predictors that were available from the DCRA database included demographic information (age, gender, body mass index (BMI)), tumour stage and location, American Society of Anesthesiologists (ASA) score and comorbidity. Comorbidity included previous abdominal surgery, cardiac comorbidity (including arrhythmias, myocardial infarction, cardiac surgery and cardiomyopathy), pulmonary comorbidity (COPD/asthma/emphysema and other) and previous thromboembolic conditions such as pulmonary embolism (PE) or deep venous thrombosis (DVT). From the comorbidity data, a Charlson Comorbidity Index (CCI) was calculated [27]. 

From the EMR, the following preoperative additional candidate predictors were extracted: undernutrition (or at risk of becoming undernourished), functional impairment, the use of a mobility aid (the use of a cane, crutches, a walking frame or wheelchair) and the risk of delirium and falls in the past 6 months. Functional impairment was assessed with the six-item Katz ADL [28] consisting of questions regarding bathing, dressing, using the toilet, eating, transferring from bed to chair and the use of incontinence materials. Risk for delirium was assessed using three yes or no questions concerning a previous delirium during hospitalisation, self-reported need for ADL assistance (in the past 24 h) and self-reported cognitive impairment. Polypharmacy (using five or more prescribed medications) was based on preoperative medication/prescriptive data from the EMR. All predictors from the EMR had been registered on the day of hospital admission or in the weeks before surgery (up to 6 weeks).

### 2.4. Statistical Analysis

Data were inspected for missing variables. Missing predictor data were estimated in a regression model using all other predictor variables and outcomes as independent variables. Missing data on candidate predictors were subsequently imputed with a single imputation technique and used for final predictor selection and model development.

Baseline characteristics were reported as means with standard deviation (SD) or as frequencies and percentages. Before imputation, candidate predictors were related to the outcome using univariable logistic regression analysis to estimate odds ratio (OR) with corresponding 95% confidence interval (CI) and *p*-value. 

To investigate the added value of geriatric predictors, two models were created. A “demographic model” included only preoperative demographic predictors, comorbidity, tumour location and stage and ASA score. For a “geriatric” model, the geriatric predictors from the EMR were added to all candidate predictors from the demographic model. 

The questions of the Katz ADL, self-reported need for ADL assistance, previous delirium and self-reported cognitive impairments (classified as a risk for delirium) were added as a categorical predictor on an individual level and dichotomised (Katz ADL ≥2 and risk for delirium ≥1). Because of expected collinearity between Katz ADL questions and the self-reported need for ADL assistance, either the Katz ADL or self-reported ADL assistance was used as a candidate predictor.

In both the demographic and geriatric model, the final model selection was obtained using the least absolute shrinkage and selection operator (LASSO) method. LASSO applies a penalty on the absolute value of the regression coefficients, such that some are set to zero, whereas others are shrunk towards smaller (absolute) values. Variables that are shrunk to zero are omitted from the model. The goal of this process is to minimise the prediction error. Compared to backward selection, the addition of shrinkage may improve model performance by avoiding overfitting and miscalibration [29]. 

The validity of both models was tested by performing bootstrap validation with 500 replications and optimism correction. The discriminative predictive performance of the models was demonstrated with the area under the curve (AUC). For the optimism corrected model, no valid 95% CI can be calculated. The final shrunk coefficients from the LASSO were used to generate a score chart, which is intended as a clinical tool. The shrunk β coefficients from the geriatric model were rounded for selection in the simplified clinical tool. Predictors with a β of less than 0.1 were therefore not selected for the clinical tool to increase the robustness of the model [29]. At least 1 point was given to each predictor included. Subsequent risk groups were created based on at least 70 observations in each risk category. 

Imputation, LASSO shrinkage and bootstrap validation were analysed with R, version 3.5.2 (R Foundation for Statistical Computing, Vienna, Austria) using “mice”, “rms” and “glmnet” packages. All other analyses were performed using SPSS version 23.0 (SPSS, Inc., Chicago, IL, USA).

## 3. Results

### 3.1. Participants

The total cohort consisted of 1366 older patients who underwent colorectal resection between January 2014 and December 2017 (Figure 1). From one hospital, data were only available from January 2014 until December 2015 because of a change in EMR registration. There were no missing demographic data (Table 1). The number of complete cases was 977 (89.8%); 87 cases (8%) had one missing candidate predictor, and 24 (2%) had two or more missing candidate predictors. Mean age was 77.7 (SD 5.2); there were 498 (46%) females, 270 (25%) patients with rectal cancer and 354 (33%) patients with an ASA score of III or IV. 

### 3.2. Model Development

There were 171 patients (16%) with one or more severe complications recorded; 51 patients were admitted to the ICU for more than two days, 26 of whom had a reintervention. A total of 121 patients (including 29 ICU patients) had a hospital stay of >14 days; 30-day mortality was 1.7% (*n* = 19). The distribution of severe complications is available in the Appendix A (Table A1).

Unadjusted associations between each candidate predictor and severe complications are shown in Table 1 and the Appendix A (Table A2). For the demographic model development, with only demographic candidate predictors, the final predictors were age, gender, COPD/asthma/emphysema, previous PE or DVT, ASA score and tumour location. The AUC of the demographic model was 0.65 (95% CI 0.62–0.70), which was corrected to AUC 0.62 after internal validation.

The discriminatory performance of the preoperative model improved to 0.69 (95% CI 64–0.73) when the geriatric predictors delirium, cognitive impairments, ADL assistance, the use of mobility aid and polypharmacy were included. The optimism corrected AUC was 0.65. Table 2 shows the regression coefficients of the demographic and geriatric models. 

When the predictors Katz ADL (instead of self-reported ADL assistance) and risk for delirium (score ≥1) were included as candidate predictors in the geriatric model, this yielded an AUC of 0.69 (95% CI 0.65–0.73) after internal validation and an AUC of 0.65 in the optimism corrected model. Judged by its clinical applicability, we used the first model (with self-reported ADL assistance) for further risk score development.

### 3.3. Clinical Prediction Model

For the development of a clinically useful prediction model and tool, the regression coefficients from the geriatric model were used to develop the geriatric colorectal cancer model (GerCRC). After rounding, age (every 10 years, b = 0.04), ASA score (b = 0.02) and self-reported cognitive impairment (b = 0.09) were omitted due to their marginal effect (b < 0.1). 

Based on the weight of the regression coefficients, all predictors were given 2 points except for tumour location (1 point) and the (self-reported) need for ADL assistance (1 point). In the simplified model, a total of 14 points can be obtained (Table 3). The number of patients with a score of 0 was 171 (16%), and the number of patients with a score of 1 was 46 (4%). The maximum score obtained by patients in our study was 11; this score was obtained for three patients, of whom two (67%) had a severe complication. 

After grouping patients with a score of 0–1 and 7 or higher, Table 4 shows the corresponding predicted proportion of complications with corresponding sensitivity and specificity. At a score of 5, the difference between predicted risk and observed risk was 6% (19% versus 13%); at a score of 7, this was 14% (31% versus 45%, respectively).

## 4. Discussion

This study set out to establish what factors are associated with severe postoperative complications after CRC surgery in order to develop a preoperative clinical prediction model for older patients. Based on tumour and preoperative registry and geriatric data of 1088 patients, the use of a mobility aid, risk factor for delirium and polypharmacy were identified as strong and important predictors for severe complications after surgery for CRC. Adding geriatric predictors to demographic and tumour-related predictors improved the model’s prognostic accuracy for older patients. With an AUC of 0.65 after optimism correction, stronger predictions are needed for better discrimination.

Gender, COPD/asthma/emphysema, previous PE or DVT, rectal cancer, previous delirium, self-reported need for ADL assistance and polypharmacy were selected as predictors to develop the GerCRC clinical prediction model. Gender, rectal cancer and severe comorbidity are well-known predictors for poor outcomes of colorectal surgery, also in older patients [26]. We recently showed strong associations between ADL and postoperative complications [13] in line with other studies in older CRC and non-CRC patients [30,31,32]. A recent geriatric pilot of the ACS-NSQIP among orthopaedic and vascular surgery patients also identified physical functioning, the use of a mobility aid preoperatively and cognitive functioning as important predictors for 20 of the 25 outcomes measured [20]. For polypharmacy and postoperative outcomes, results have been conflicting [33].

In contrast to other prediction models for mortality, anastomotic leakage or surgical site infections [9,11,15,34,35], age and ASA score had no additional predictive value in our study. This is in accordance with a study among older patients with CRC referred for GA [32]. Several explanations can be put forward. First, because our study population was limited to older patients, the age distribution is smaller and therefore less likely to be discriminative. Possibly, in our model, calendar age (and possibly ASA score) were replaced by measures of age-related problems such as cognitive functioning, functional performance and comorbidity. Second, in the Netherlands, national guidelines recommend geriatric screening of older patients scheduled for CRC surgery to identify high-risk surgical patients and guide interventions or adapt treatment plans. This means our study population could be somewhat selected, as we have no information on the nonsurgically treated older patients in our cohort. 

After interval validation, the expected discrimination of our model was 0.65. Because we aimed to develop a model that can be used in preoperative decision-making, we did not include perioperative predictors such as the surgical technique (laparoscopic surgery or not) or complications. Moreover, high-risk patients such as patients with metastatic disease or acute surgery [4] were not included. When these predictors and patients were added, the GerCRC model performance improved (data not shown). When our GerCRC model is externally validated, more focus will be placed on the calibration of the different risk groups to judge the performance and clinical usefulness of this model [36].

A head-to-head comparison with the POSSUM [14], CR-BHOM [18] and ACS-NSQIP original and recently published universal model [10,19] should be made with caution because of differences in the definition of severe complications, the use of perioperative predictors and the lack of external validation. The GerCRC model is the only model that uses a prolonged length of hospital stay in the definition of a severe complication, accounting for a possible negative impact of a prolonged hospital stay on physical functioning and quality of life. The use of perioperative predictors in the other models limits preoperative decision-making.

External validation for all models (including the GerCRC model) has not been performed or was shown to be somewhat disappointing for older patients. The POSSUM was shown to overpredict complication and mortality risk. A recent evaluation of the performance of the POSSUM in 1380 U.K. patients (with surgery between 2008 and 2013) confirmed its poor discriminatory performance for severe complications (AUC 0.51) [37]. The discriminatory performance for prediction morbidity in 204 Portuguese octogenarians was 0.65 for the POSSUM and 0.66 for the CR-BHOM model with poor calibration [38]. The original ACS-NSQIP surgical risk model was not specifically developed for colorectal cancer surgery. Moreover, the accuracy of the universal ACS-NSQIP model for severe complications and its performance for outcomes in older CRC patients have not been published. The accuracy of the universal ACS-SNQIP model for severe complications in 200 older gynecologic oncology patients undergoing laparotomy (2009–2013) was only 0.62, also with poor calibration [39]

To account for possible heterogeneity between cohorts [29], external validation of the proposed prediction models is required. Changes in the healthcare setting and geographic differences are also reasons for periodic updating and recalibration [40]. That concerns both the ACS-NSQIP model, which has not been validated outside the United States, and the GerCRC model. A more detailed comparison of the preoperative GerCRC, CR-BHOM and ACS-NSQIP models is shown in the Appendix A (Table A3).

The strengths of our study are the reasonable sample size of high-quality prospectively collected data, the inclusion of geriatric predictors and the use of statistical techniques to take into account possible optimism. A limitation of our study is the relatively low number of index events in our model development. Only 16% of the patients experienced a severe complication. With 19 candidate predictors, the 10:1 ratio was exceeded, which is suggested to decrease the risk of selecting noise predictors [41]. However, no previous unknown predictors were selected. We further note that self-reported physical function can be overestimated in some older patients [42]. In addition, in the present study, patients were only included in the analysis when surgery was performed. Hence, the results do not apply to patients who were considered too frail for surgery. However, the Netherlands cancer registry shows that only 5% of patients with colon cancer over age 70 and 20% of rectal cancer patients do not receive surgical treatment [43]. 

Providing accurate prognostic information to older CRC patients concerning the possible risks and benefits of their surgical treatment is important for several reasons. Prediction tools enable discussing risks of adverse treatment outcomes with a potential negative effect on quality of life and physical functioning [44] and improve the likelihood that treatment decisions are consistent with the needs, values and preferences of patients. Furthermore, they can direct alternative treatment options when available, and, finally, when high-risk populations can be identified, interventions aimed to improve surgical outcomes may become feasible. Therefore, the pre-CRC model has good potential to be used for preoperative decision-making, providing better and more accurate estimates of the risk of surgery. 

Possible future research could study whether outcomes of surgery improve when predictors of complications such as low physical functioning and pulmonary comorbidity are corrected or improved by preoperative interventions such as prehabilitation [45], pulmonary optimisation [46] and geriatric comanagement [47]. It is possible that the GerCRC model can be improved in the future by adding information on cognitive functioning.

## 5. Conclusions

The GerCRC is the first prediction model specifically developed for older patients expected to undergo CRC surgery. Combining tumour and geriatric predictors in the GerCRC model modestly improves performance in the heterogeneous older population. After external validation, this risk model could serve as a basis for preoperative decision-making.

## Figures and Tables

**Figure 1 cancers-13-03110-f001:**
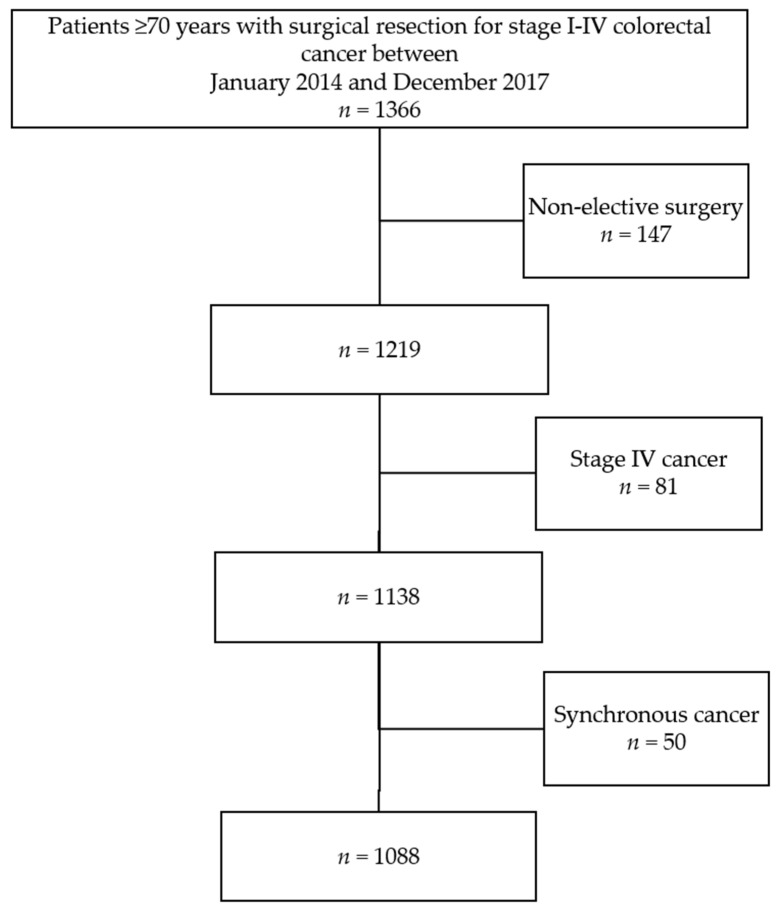
Patient selection.

**Table 1 cancers-13-03110-t001:** Baseline characteristics and univariable associations with severe complications.

		No Patients = 1088			Odds Ratio (95% CI)	
	Missing	All	Severe Complication			
Predictors			Yes	No		*p*–Value
Demographics						
Age Years (mean and SD)	–	77.67 (5.2)	78.5 (5.2)	77.51 (5.1)	1.038 (1.01–1.07)	0.017
Age Categories						
70–74	–	383 (35)	47 (12)	336 (88)	reference	
75–79	–	353 (32)	62 (18)	291 (82)	1.52 (1.01–2.30)	0.044
80–84	–	241 (22)	40 (17)	201 (83)	1.42 (0.90–2.25)	0.13
85+	–	111 (10)	22 (20)	89 (80)	1.77 (1.01–3.09)	0.045
Gender	–					
Females	–	498 (46)	60 (12)	439 (88)	reference	
Males	–	590 (54)	111 (19)	479 (81)	1.69 (1.2–2.38)	0.002
BMI kg/m^2^ (mean and SD)	–	26.48 (11.4)	26.8 (4.4)	26.4 (12.3)	1 (0.99–1.01)	0.71
BMI Categories						
<25 kg/m^2^	–	464 (43)	62 (13)	402 (87)	reference	
25–30 kg/m^2^	–	467 (43)	81 (17)	386 (83)	1.36 (0.95–1.95)	0.09
>30 kg/m^2^	–	157 (14)	28 (18)	129 (82)	1.41 (0.86–2.29)	0.17
Comorbidity						
History of Abdominal Surgery	–	460 (42)	75 (16)	385 (84)	1.08 (0.78–1.50)	0.65
Cardiac Comorbidity	–	401 (37)	74 (18)	327 (82)	1.38 (0.99–1.92)	0.06
COPD/Asthma/Emphysema	–	110 (10)	30 (27)	80 (73)	2.27 (1.41–3.51)	0.001
Previous PE or DVT ^a^	–	52 (5)	15 (29)	37 (71)	2.56 (1.25–4.44)	0.008
Charlson Comorbidity Index (median and range)		1 (0–2)	1 (0–8)	1 (0–7)	1.27 (1.03–1.56)	0.022
Comorbidity CCI ≥ 2	–	392 (36)	76 (19)	318 (81)	1.49 (1.07–2.07)	0.02
ASA Score (mean and SD)		2.3 (0.6)	2.4 (0.7)	2.2 (0.6)	1.61 (1.24–2.07)	<0.001
I–II	–	734 (67)	97 (13)	637 (87)	reference	
III–IV	–	354 (33)	74 (21)	280 (79)	1.74 (1.24–2.42)	0.001
Tumour Location						
Colon	–	818 (75)	120 (15)	698 (85)	reference	
Rectum	–	270 (25)	51 (19)	219 (81)	1.35 (0.94–1.94)	0.099
Tumour Stage						
I	–	336 (31)	54 (16)	282 (84)	reference	
II	–	411 (38)	63 (15)	348 (85)	0.95 (0.64–1.04)	0.78
III	–	341 (31)	54 (16)	287 (84)	0.98 (0.65–1.48)	0.93
Surgical Approach						
Laparoscopic	–	877 (81)	119 (14)	758 (86)	reference	
Open	–	211 (19)	52 (25)	159 (75)	2.08 (1.44–3.01)	<0.001
Geriatric						
Katz ADL (mean and SD)	15	0.3 (0.8)	0.5 (1.3)	0.2 (0.7)	1.38 (1.18–1.61)	<0.001
score ≥2		65 (6)	22 (34)	43 (66)	2.97 (1.73–5.11)	<0.001
Reported Falls	76	129 (12)	24 (19)	105 (81)	1.19 (0.74–1.92)	0.47
Risk for Malnutrition	12	215 (20)	37 (17)	156 (73)	1.35 (0.90–2.02)	0.1
Risk for Delirium (mean and SD)	18	0.3 (0.6)	0.5 (0.8)	0.2 (0.6)	1.69 (1.34–2.12)	<0.001
Delirium Score ≥1		210 (19)	56 (27)	154 (73)	2.38 (1.65–3.42)	<0.001
Medication Use (mean and SD)	18	4 (0–17)	5 (0–17)	4 (0–16)	1.1 (1.05–1.56)	<0.001
Polypharmacy (No. ≥5)		490 (45)	103 (21)	387 (79)	2.18 (1.55–3.07)	<0.001
Preoperative Use of a Mobility Aid	21	191 (18)	51 (27)	116 (61)	2.39 (1.64–3.47)	<0.001

Mean with Standard Deviation (SD). Median with (range) and frequencies with percentage (%). Odds Ratio’s with 95% Confidence Interval (CI). ^a^ PE, Pulmonary Embolism; DVT, Deep Venous Thrombosis.

**Table 2 cancers-13-03110-t002:** Model development and multivariable regression coefficients after shrinkage.

	Demographic Model	Geriatric Model
Predictors	Beta ^a^	Beta ^a^
Cohort model estimates		
Intercept	−6.64	−2.64
Age (for every 10 years)	0.14	0.04
Male gender	0.26	0.32
BMI, kg/m^2^	–	–
History of abdominal surgery	–	–
Cardiac comorbidity	–	–
COPD/asthma/emphysema	0.27	0.34
Previous PE or DVT ^b^	0.37	0.35
ASA score	0.2	0.02
Rectal tumour	0.03	0.12
Tumour stage	*	–
Reported falls	*	–
Risk for malnutrition	*	–
Previous delirium	*	0.33
Self-reported cognitive impairment	*	0.09
Self-reported need for ADL assistance	*	0.16
Mobility aid	*	0.43
Polypharmacy (≥5)	*	0.35
Model performance (AUC)		
Model after bootstrapping	0.648	0.687
Optimism corrected model	0.623	0.650

^a^ Regression coefficient after shrinkage using LASSO. – candidate predictor was not selected after shrinkage. * candidate predictor was not used in model development. ^b^ PE, Pulmonary Embolism; DVT, Deep Venous Thrombosis.

**Table 3 cancers-13-03110-t003:** Geriatric colorectal cancer model (GerCRC) score chart.

Characteristic	Score
Male gender	2
COPD/asthma/emphysema	2
Previous PE or DVT ^a^	2
Rectal cancer	1
Mobility aid	2
Previous delirium	2
Need for ADL assistance	1
Polypharmacy	2
Total Score (add all)	
Probability of developing a severe complication (Table 3)	%

^a^ PE, pulmonary embolism; DVT, deep venous thrombosis.

**Table 4 cancers-13-03110-t004:** Probability of severe complications after CRC surgery in relation to the sum score from Table 3.

Score from Table 3	Events/No. Cases	Predicted	Sensitivity ^a^	Specificity ^a^	+LR ^b^	−LR ^b^
0–1	18/217	10%	1	0	1	-
2	28/293	13%	0.89	0.22	1.14	0.49
3	20/139	14%	0.73	0.51	1.48	0.53
4	37/198	17%	0.61	0.64	1.69	0.61
5	11/86	19%	0.40	0.81	2.11	0.74
6	23/80	23%	0.33	0.89	3.12	0.75
7-or higher	34/75	31%	0.20	0.96	4.45	0.84

^a^ Sensitivity and specificity based on the development cohort. ^b^ LR, likelihood ratio; +, positive; −, negative.

## Data Availability

None.

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
