# Peer review of "A Prediction Model for Severe Complications after Elective Colorectal Cancer Surgery in Patients of 70 Years and Older"

_cancers, 2021, doi:10.3390/cancers13133110_

Round 1
Reviewer 1 Report
This multicenter study aims to evaluate the added value of some parameters of multidimensional geriatric assessment in terms of prediction of severe complication after CRC surgery.
This is a retrospective study of prospectively collected data.
The high number of patients included is certainly a strength element.
The study is well written and structured.
However, there are some criticisms and weak elements.
1) line 106: the authors state that ERAS represented the standard of care. How many items were followed? All? This data is not specified and could have greatly influenced the results.
2) line 108: the authors state that they have assessed the risk of malnutrition. Was malnutrition treated prior to surgery? It does not appear correct to plan elective surgical treatment in a malnourished patient.
Furthermore, the authors talk about post-surgical rehabilitation in patients with compromised ADLs. Wouldn't it be more correct to plan pre-habilitation to reduce the operative risk?
3) line 115: the main outcome is the evaluation of "severe complications". However, the authors define "severe complication" in generic and arbitrary way. An objective rating scale is necessary, for example the Clavien-Dindo classification. This is a critical element of the work. Furthermore, the rate of "severe complications" and "mortality" are extremely low (16% and 2% respectively), especially in an elderly patient group, to obtain convincing statistical results.
4) Among the assessment tools there is no reference to the evaluation of cognitive status, for example through the MMSE or SPMSQ, which are considered fundamental tools for geriatric assessment.
5) Line 122: how have the co-morbidities been evaluated and classified? how were they included in the final predictive model? An evaluation tool such as CIRS (CUMULATIVE ILLNESS RATING SCALE) would seem more appropriate.
6) the authors state (line 276-280) that all patients underwent geriatric screening before being enrolled for surgical treatment: this may represent a selection bias for the study since frail patients could be excluded.
7) The GerCRC does not include the planned surgical approach. It is difficult to believe that the laparotomic or laparoscopic approach have the same impact on the outcome. In univariate evaluation, laparoscopy appears to have the best results.
Author Response
We thank the reviewer very much for the time and effort put in reading and commenting on our manuscript. The text of the original manuscript has been modified in line with the valuable comments and recommendations of the reviewer.
Reviewers' comments/questions (Q) with our rebuttal (R):
Q1. line 106: the authors state that ERAS represented the standard of care. How many items were followed? All? This data is not specified and could have greatly influenced the results.
Unfortunately, we do not have information on the compliance to the Enhanced Recovery After Surgery (ERAS) protocol for individual patients or for participating hospitals. We agree that there could have been possible heterogeneity in the extent to which ERAS-guidelines were followed in different hospitals. However, in all participating hospitals both laparoscopy and ERAS (anesthesiology guidelines and postoperative care) were considered the standard of care and therefore most likely used in the majority of patients.
The following sentence has been changed following this comment and reads as follows:
“In all participating hospitals, both laparoscopy and The Enhanced Recovery After Surgery (ERAS, anesthesiology guidelines and postoperative care) [24] were considered the standard of care and therefore most likely used in the majority of patients.” (lines 107-109).
Q2: line 108: the authors state that they have assessed the risk of malnutrition. Was malnutrition treated prior to surgery? It does not appear correct to plan elective surgical treatment in a malnourished patient.
We thank the reviewer for this excellent remark. Detection of possible malnutrition is part of ERAS and preoperative care for all patients. The risk of malnutrition is assessed preoperatively with the screening tools Short Nutritional Assessment Questionnaire (SNAQ) [1] or Malnutrition Universal Screening Tool (MUST).[2] Additional information is collected on high-risk patients using a more comprehensive intake with additional nutritional interventions by a dietician when needed (line 109).
The following lines were added to the method section (lines 114-117): “The risk of malnutrition is assessed preoperatively with the screening tools Short Nutritional Assessment Questionnaire (SNAQ) [1] or Malnutrition Universal Screening Tool (MUST).[2] Additional information is collected on high-risk patients using a more comprehensive intake with additional nutritional interventions by a dietician when needed.”
Q3. authors talk about post-surgical rehabilitation in patients with compromised ADLs. Wouldn't it be more correct to plan pre-habilitation to reduce the operative risk?
We agree with the reviewer that prehabilitaiton is a possible strategy for for improving postoperative outcome. This section now reads as follows: “Possible future research could study whether outcomes of surgery improve when predictors of complications such as low physical functioning and pulmonary comorbidity are corrected or improved by preoperative interventions such as prehabilitation [45], pul-monary optimisation [46], and geriatric co-management [47]. It is possible that the GerCRC model can be improved in the future by adding information on cognitive func-tioning.”(lines 340-345).
Q4. line 115: the main outcome is the evaluation of "severe complications". However, the authors define "severe complication" in generic and arbitrary way. An objective rating scale is necessary, for example the Clavien-Dindo classification. This is a critical element of the work. Furthermore, the rate of "severe complications" and "mortality" are extremely low (16% and 2% respectively), especially in an elderly patient group, to obtain convincing statistical results.
The reviewer makes an interesting suggestion. Unfortuanately, the Dutch Colorectal Audit (DCRA) does not provide this information for our cohort of patients. Any complication is registered but no classification according to Clavien Dindo is made for each complication. Our definition of a servere complication is consistent with previous publications in which data from the DSCA were analysed.[3] We agree this has limitations but is in line with a more comprehensive definition of servere compliations used by other prediction models (Appendix C). In addition, older patients with a surgical or non-surgical complication and a hospital stay of more then 14 days most likely to influence outcomes such as quality of life and retaining the independence [4].
Despite the low proportion of severe complications, we found a significant and strong correlation between predictors and outcome. The low number of complications is in line with a recent publication on outcomes of all patients included in the DCRA between 2009 and 2016.[5]
Q5. Among the assessment tools there is no reference to the evaluation of cognitive status, for example through the MMSE or SPMSQ, which are considered fundamental tools for geriatric assessment.
No full geriatric assessment of full cognitive evaluation is performed for all patients in our hospitals, but 2 screening questions were asked (self reported cognitive functioning en previous delirium). It is possible that the model can be improved in the future by adding information on cognitive functioning.
The following sentence was added to the manuscript: “It is possible that the model can be improved in the future by adding information on cognitive functioning.” (line 345-346)
Q6. Line 122: how have the comorbidities been evaluated and classified? how were they included in the final predictive model? An evaluation tool such as CIRS (CUMULATIVE ILLNESS RATING SCALE) would seem more appropriate.
When a composite predictor such as CIRS is used in risk prediction models, crucial prognostic information is lost because this requires defining thresholds or cut-offs [6]. Therefore, we decided not to use CIRS and instead identified individual comorbidities associated with poor outcomes of colorectal cancer surgery from the literature.
Q7. The authors state (line 276-280) that all patients underwent geriatric screening before being enrolled for surgical treatment: this may represent a selection bias for the study since frail patients could be excluded.
We thank the reviewer for this comment en suggestion. We agree with the reviewer and added to the discussion: “Lastly, in the present study patients were only included in the analysis when surgery was performed and hence the results do not apply to patients who were considered too frail for surgery. The Netherlands cancer registry shows however that only 5% of patients with colon cancer over age 70 and 20 % of rectal cancer patients do not receive surgical treatment. [7] (lines 326-330)
Q8. The GerCRC does not include the planned surgical approach. It is difficult to believe that the laparotomic or laparoscopic approach have the same impact on the outcome. In univariate evaluation, laparoscopy appears to have the best results.
Indeed. Because we wanted to develop a preoperative model, we only included preoperative predictors. Our model improved when the peri-operative preditor laparoscopic surgery was included (AUC +/0.72-0.75). We decided to exclude this predictor and accept a lower discriminatory power of our model, thereby retaining clinical applicability.
Following this comment, this part of the discussion now reads as follows “we did not include peri-operative predictors such as the surgical technique (laparoscopic surgery or not) or complications“ (line 284-285)
References
- Kruizenga, H.M.; Seidell, J.C.; de Vet, H.C.; Wierdsma, N.J.; van Bokhorst-de van der Schueren, M.A. Development and validation of a hospital screening tool for malnutrition: the short nutritional assessment questionnaire (SNAQ). Clin. Nutr. 2005, 24, 75-82, doi:10.1016/j.clnu.2004.07.015.
- Cawood, A.L.; Elia, M.; Sharp, S.K.; Stratton, R.J. Malnutrition self-screening by using MUST in hospital outpatients: validity, reliability, and ease of use. Am. J. Clin. Nutr. 2012, 96, 1000-1007, doi:10.3945/ajcn.112.037853.
- Henneman, D.; Snijders, H.S.; Fiocco, M.; van Leersum, N.J.; Kolfschoten, N.E.; Wiggers, T.; Wouters, M.W.; Tollenaar, R.A. Hospital variation in failure to rescue after colorectal cancer surgery: results of the Dutch Surgical Colorectal Audit. Ann. Surg. Oncol. 2013, 20, 2117-2123, doi:10.1245/s10434-013-2896-7.
- Abeles, A.; Kwasnicki, R.M.; Pettengell, C.; Murphy, J.; Darzi, A. The relationship between physical activity and post-operative length of hospital stay: A systematic review. International Journal of Surgery 2017, 44, 295-302, doi:https://doi.org/10.1016/j.ijsu.2017.06.085.
- Henneman, D.; Ten Berge, M.G.; Snijders, H.S.; van Leersum, N.J.; Fiocco, M.; Wiggers, T.; Tollenaar, R.A.; Wouters, M.W. Safety of elective colorectal cancer surgery: non-surgical complications and colectomies are targets for quality improvement. J Surg Oncol 2014, 109, 567-573, doi:10.1002/jso.23532.
- Steyerberg, E.W.; Uno, H.; Ioannidis, J.P.A.; van Calster, B. Poor performance of clinical prediction models: the harm of commonly applied methods. J. Clin. Epidemiol. 2018, 98, 133-143, doi:10.1016/j.jclinepi.2017.11.013.
- Netherlands Cancer Registry (NKR); [Accessed 01 June 2021]: Retrieved from www.cijfersoverkanker.nl.
Reviewer 2 Report
The authors developed and internally validated a prognostic preoperative clinical model for severe postoperative complications after elective surgery for colorectal cancer, intended to support shared decision making with older patients. The article is well structured with well conducted statistical analysis, including up to 1088 patients. The discussion is very well conducted and correctly underlines possible differences with other studies and the limits of the present manuscript. I congratulate with the authors for the very good job, and I am fully in support of the publication of their paper. AcceptAuthor Response
We thank the reviewer for the time and effort put in reading and commenting on our manuscript.
Reviewer 3 Report
Dear authors,
This is a well written study about prediction of severe postoperative complication after colorectal surgery for elderly (>70 y.o.) patients. It is clinically important to know the risk of surgery for the elderly in the recent rapidly aging society. I think it logical and important that the authors created the prediction model using only preoperative information.
I don’t agree with the idea of including “postoperative stay >14 days” in the severe complications. Furthermore, this is the most common factor among severe complications according to the appendix A. I don’t think that physicians give up surgery if the patients will stay long because of, for example, delirium or postoperative ileus. Whether the condition can be fully recovered or not is the matter of interest. In that sense, decrease in ADL score after surgery might be the candidate for severe complication.
Though the AUC (0.69 and corrected value of 0.65) is relatively low, I think the result is still meaningful, since the data came from well-designed multi-institutional study group.
Some minor comments are as follows.
In page 8, the table 2, “ASTMA/emfysema” should be “Asthma/emphysema”?
In page 10 the table 3A, patients with “mobility aid” inevitably “need ADL assistance”. Is there any meaning of discriminating them?
In page 10 line 243 to 244 should be deleted?
Regards,
Author Response
In page 8, the table 2, "ASTMA/emphysema", should be "Asthma/emphysema"? In page 10, line 243 to 244 should be deleted?
R1. We thank the reviewer for the time and effort put in reading and providing comments to improve the manuscript. The suggested changes have been made.
Q2. On page 10 the table 3A, patients with "mobility aid" inevitably "need ADL assistance". Is there any meaning of discriminating them?
R2. The number of patients needing ADL assistance is much lower than the number of patients that use a mobility aid. As shown, the strongest predictor is the use of a mobility aid. Adding ADL assistance improved the model further (beta >0.10). Because this is a cumulative score, including both showed superior discrimination compared to including them separately.
Round 2
Reviewer 1 Report
I thank the authors for having accurately answered all the questions I raised above. Now the text is certainly improved in understanding and presentation of the results.